# IMBALANCE: INFERENCE-TIME LATENT SEARCH AGAINST DEGREE IMBALANCE IN LINK PREDICTION

## ABSTRACT

Knowledge Graph Embedding models have been extensively used to learn representations of entities and relations in Knowledge Graphs for predicting missing links. However, the quality of the learned representations varies a lot across different areas of the same Knowledge Graph. If previous research efforts have loosely linked the problem to relation types or degree bias, we show that it is much more widespread, and it more precisely lies in the degree imbalance of the entities in test triples. In particular, we show that the prediction of a target entity that has a degree much smaller than the degree of the anchor entity is extremely problematic. This is critical in use cases like *drug target discovery*, where these triples are predominant, or *recommender systems*, where they represent important corner cases. To address this issue, we propose an inference-time latent search optimization method capable of significantly improving model predictions on the most imbalanced triples. Built on top of a pre-trained model, it explores the embedding space at evaluation time, blending known and out-of-band information to mitigate the degree imbalance bias. We show the value of our approach on imbalanced triples from common benchmark datasets, where we outperform conventional methods, opening the door to the successful adoption of Knowledge Graph Embedding models on these critical corner cases.

## 1 INTRODUCTION

Knowledge Graphs (KGs) are flexible and scalable data structures that model factual knowledge by linking concepts through relationships (Hogan et al., 2021). The possibility of representing and integrating virtually any type of knowledge has made them scale up to include millions or billions of facts. However, such a size is incompatible with manual curation and leaves the door open for incompleteness (Dong et al., 2014). In this scenario, the design of a machine-based approach to infer missing links has attracted a lot of attention from the scientific community. This task is known as *link prediction*, where the goal is to predict whether a relation $p$ exists between two entities $s$ and $o$ in the graph. In this context, a *query* takes the form of an incomplete triple, such as $(s, p, ?)$ or $(?, p, o)$ and the model must infer the missing entity. The entity that is already given in the query (e.g., the subject in $(s, p, ?)$) is referred to as the *anchor node*, since predictions are conditioned on it. The most successful attempts to solve this task have been carried out through Knowledge Graph Embedding (KGE) models, a family of scalable methods that learn low-dimensional representations for entities and relations.

Despite their success, KGEs suffer from limitations, as the quality of the learned representation significantly varies across the graph, strongly limiting the model performance under certain conditions. Previous work has attributed such behavior to relation types (Bordes et al., 2013; Wang et al., 2014; Lin et al., 2015; Ji et al., 2015; He et al., 2015) and degree bias (Mohamed et al., 2020; Shomer et al., 2023). In this work, we provide a more precise analysis, identifying the root cause of the problem in the *degree imbalance* of the head and tail entities in test triples. We show how the bigger the degree difference between the two, the poorer the prediction of the lower degree entity and the better the prediction of the higher degree entity. We further tie the problem to a two-fold learning issue that affects low-degree entities, highlighting how strong overfitting and failed convergence lie behind unsuccessful predictions.

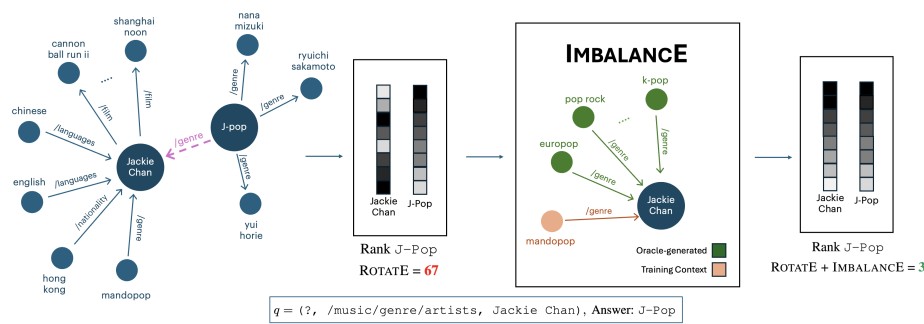

**Figure 1:** We consider a query $q$ with high degree imbalance. In this example taken from FB15k-237, `Jackie Chan` has degree 49 and `J-Pop` has degree 8 (limited in the figure for clarity of visualization). IMBALANCE takes the pre-trained embeddings of the KGE model for the anchor entity and performs an inference-time latent search optimization on a subset of the original neighborhood enhanced with the triples generated by the oracle. It then outputs a representation for the anchor node that better aligns with the selected query, improving the ranking of the target answer from 67 down to 3.

For example, in drug discovery a key task is identifying proteins associated with a disease of interest. When framed as a link prediction task, where KG nodes are biological entities (diseases, proteins, genes. . . ), top-scored proteins returned as solutions to $q = (?protein, associatedWith, disease)$ represent promising associations. These target triples are often imbalanced, as we typically have much more information (i.e., a much higher degree) for conditions as opposed to proteins (biologists have only identified a relatively small fraction of proteins with sufficient detail). Degree imbalance compromises our ability to successfully predict meaningful solutions, undermining the reliability of link prediction in this critical application.

Another example is *recommender systems*, where link prediction has been deployed frequently (Zhang et al., 2016). On a music streaming platform, we want to complement the information about artists as much as possible. In this way, the recommendations to accommodate user tastes are going to be more accurate. Consider (`J-pop`, `/music/genre/artists`, `Jackie Chan`), a test triple from FB15k-237 (Toutanova & Chen, 2015). The degree of `J-pop` is low, while `Jackie Chan` has a much higher number of connections. We want to answer the query $q = (?, /music/genre/artists, Jackie Chan)$, so to tag `Jackie Chan` with an additional, related genre – in this case, `J-pop` – to broaden up its audience. However, when using embeddings from a pre-trained KGE model, the predicted ranking for `J-pop` is poor. This is because the large degree difference between `Jackie Chan` and `J-pop` causes the KGE model to place `J-pop` far down in the ranking (see Figure 1).

The problem affects a broad range of triples and various Graph Machine Learning (GraphML) models: from shallow architectures like TransE, DistMult, ComplEx and RotatE, to GNN-like ones, including the state-of-the-art NBFNet.

To address this issue, we propose IMBALANCE, an inference-time latent search method. Given the pre-trained embeddings of a KGE model and an imbalanced test triple $t = (s, p, o)$, in order to improve the prediction of the low-degree entity, IMBALANCE fine-tunes the embeddings of the anchor node and of few selected entities optimizing a dual-term objective function. These two terms extend the generalization of the embedding of the high-degree node and improve the quality of the embedding of the low-degree nodes. The former goal is achieved by an exploration of the embedding space guided by a selection of training facts: this second pass ring-fences the representation of the high-degree node in a region of the space suitable for the prediction. The latter goal, on the other hand, is achieved through an oracle that extends the little knowledge available in the KG for long-tail entities. Refer to Figure 1 for an example and an overview of the method.

We validate the value of our approach on the most imbalanced triples of common benchmark datasets. We show how IMBALANCE leads to significant improvements across multiple learning-to-rank metrics, enabling the prediction of low degree entities in critical applications, where these corner cases are predominant.

In summary, our work makes the following contributions:

1 We analyze the degree imbalance affecting a wide variety of KGE models, and we further tie it to a two-fold learning issue that compromises the embedding quality of low-degree entities.

2 We propose IMBALANCE, an inference-time latent search method that mitigates the above problem. We assess its impact on heavily imbalanced triples in popular *link prediction* benchmarks across multiple KGE models and show that it works as an effective plug-in to enhance KGE predictions. Moreover, we show how it can provide a lightweight and efficient alternative to much more involved and computationally expensive models.

Our paper is structured as follows: we present the related work in Section 2, then we move on to introduce the degree imbalance problem and its implications for KGE models (Section 3). In Section 4 we describe IMBALANCE, from the underlying intuition to the details of the method, while in Section 5 we present the experimental results. We finally draw conclusions and highlight limitations and future directions in Sections 6 and 7.

## 2 RELATED WORK

**Knowledge Graph Embedding Models** KGE models learn continuous representations of entities and relations in the KG from the graph topology and its soft regularities. A long list of methods (Cao et al., 2024) has followed seminal work in the space (Nickel et al., 2012). In this work, we limit our analysis to four traditional KGE models: TransE (Bordes et al., 2013), DistMult (Yang et al., 2015), ComplEx-N3 (Trouillon et al., 2016; Lacroix et al., 2018) and RotatE (Balažević et al., 2019). Despite others have claimed better performance Balažević et al. (2019), the performance gain is often marginal, and the small differences in modeling the representation of triples are not relevant to the core contribution of our work.

A notable exception is represented by NBFNet (Zhu et al., 2021), a GNN-like architecture that has achieved substantial improvements on the aggregate metrics. We will show it is nonetheless affected by the degree imbalance, but we will not include it in our main experiments. Indeed, in NBFNet the representations of different entities are heavily entangled within the message-passing architecture, which makes the direct application of our approach to this architecture not scalable and prone to overfitting.

**Topology-related Issues on KGs** The topology of KGs introduces significant challenges in training KGE modelsSardina et al. (2024). Previous work (Mohamed et al., 2020; Rossi et al., 2021) has highlighted how link prediction models have a bias for high-degree nodes and have recognized how aggregate metrics can offer a distorted view on the actual performance of models. However, they loosely defined the issue and did not propose approaches to concretely tackle these challenges. Shomer et al. (2023) looked past the degree of single nodes into frequency among entity-relation pairs and proposed the synthetic generation of additional embeddings to compensate for long-tail entities distribution. Other works (Bordes et al., 2013; Ji et al., 2015; Lin et al., 2015; He et al., 2015), on the other hand, have identified the heterogeneity of relation types (and in particular, 1-to-many, many-to-1) as one of the main obstacles. They have tried to address the issue altering the representation of relations in a more expressive way, but were not able to solve the issue. Our work surpasses all these previous efforts as it defines the problem in a more systematic way, tracing it back to a two-fold learning issue of the training and extending the scope to a much broader set of triples.

**Latent Search Optimization** The application of an inference-time latent search optimization has been carried out before in a completely different domain by Bonnet & Macfarlane (2024). To the best of our knowledge, our work is the first that leverage such a latent search optimization for KGE models. We refer to the work above for an overview of other related domains of application that are not pertinent to this work.

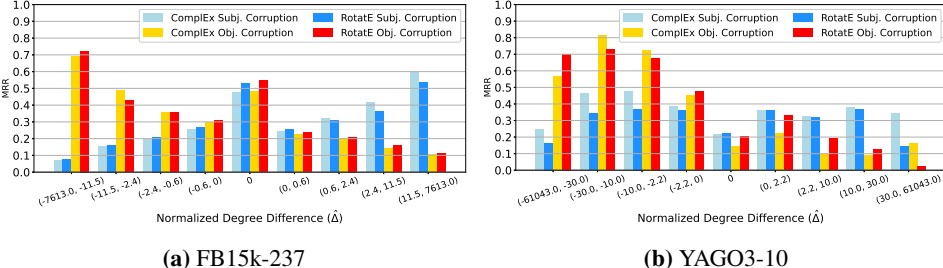

**(a)** FB15k-237          **(b)** YAGO3-10

**Figure 2:** Performance of conventional KGE models on test triples binned by normalized degree difference. Bins are obtained using quartiles of $|\hat{\Delta}(t)|$ for $t \in \mathcal{G}$. On the left hand side we have triples with high-degree object and low-degree subject, on the right hand side, high-degree subjects and low-degree objects. We isolate subject and object corruption.

## 3  WHAT AGGREGATE METRICS DO NOT SHOW

KGE models have proven to be successful in reconstructing missing links in KGs. However, what the aggregated metrics often hide is that the quality of these predictions varies hugely in different groups of triples. Previous work has loosely associated the problem with the epistemic uncertainty of low-degree entities and has shown the challenge of predicting head (and tail) of 1-to-N (and N-to-1) relations. We extend the scope of the issue, proving that the fundamental challenge in predicting the head or tail of a triple lies in the degree difference (or *degree imbalance*) between the subject and the object: this makes the reconstruction of the higher degree entity much simpler, while leaving the lower degree entity poorly reconstructed.

In Figure 2, we plot the MRR of different ComplEx and RotatE, isolating the performance on subject and object corruptions and splitting the test triples of FB15k-237 (Toutanova & Chen, 2015) and Yago3-10 (Mahdisoltani et al., 2015) based on the normalized degree difference, that we define as:

$$\hat{\Delta}(t = (s, p, o)) = \frac{\delta(s) - \delta(p)}{\min \{\delta(s), \delta(o)\}},$$

where $\delta : \mathcal{E} \to \mathbb{R}$ is the degree function, that counts the incoming and outgoing edges for $e \in \mathcal{E}$. We can immediately observe the stark difference in the quality of the predictions for highly imbalanced triples and how this difference systematically decreases as the normalized degree difference $\hat{\Delta}$ approaches zero. Despite the aggregate metrics reported in Table 1 portray models with robust predictive power, a different point of view changes things entirely: indeed, such performance is inflated by easier predictions and these models could hardly be trusted in critical use cases where the prediction of low-degree entities is the predominant goal (see Appendix A).

**Table 1:** Aggregate performance of KGE models on FB15k-237 and YAGO3-10

|  | FB15k-237 | | YAGO3-10 | |
|---|---|---|---|---|
|  | **MRR** | **H@10** | **MRR** | **H@10** |
| ComplEx | 0.31 | 0.49 | 0.36 | 0.56 |
| RotatE | 0.31 | 0.51 | 0.37 | 0.57 |

To explain the degree imbalance issue, we take a step back and look into the training process. We carry out an analysis on FB15k-237 and report the results obtained for ComplEx.

In conventional KGE models, entities and relations are assigned a low-dimensional, continuous representation to capture relational patterns in the graph structure. During every training epoch, these representations are optimized to maximize the score assigned to the ground truth triples in the KG (*positives*), and minimize the score for false statements (*negatives*). In particular, the embedding of an entity is moved around the latent space every time the model processes a positive or a negative involving that entity.

By plotting the (euclidean) distance between the embeddings of the same entity across successive epochs, we can get an idea of the magnitude of the updates and, as it decreases, of the rate of

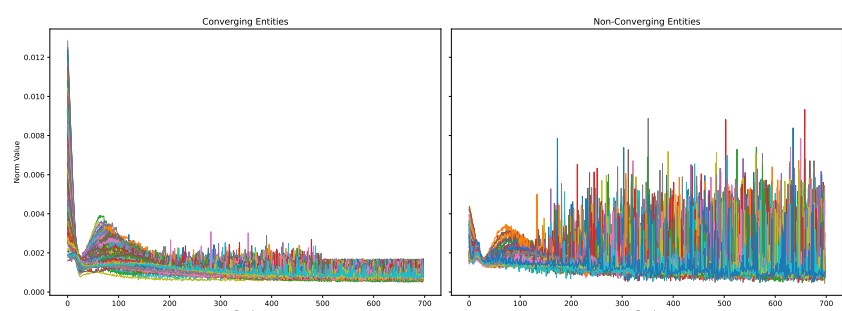

**(a)** On the left, entities converge and the update reduces to approach zero. On the right, embeddings of the entities keep getting updated, proving there is an issue with the learning process.

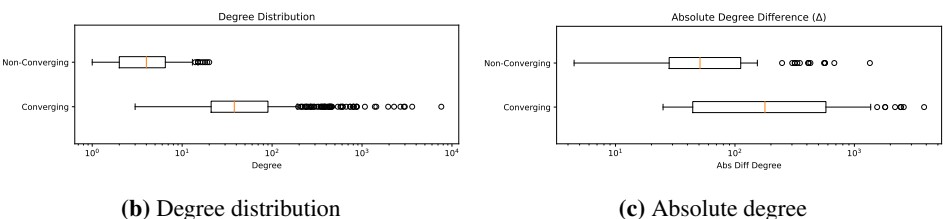

**(b)** Degree distribution

**(c)** Absolute degree

**Figure 3:** (Top) Euclidean Norm of the difference between embeddings of converging and non-converging entities while training RotatE on FB15k-237. (Bottom) Comparison of degree statistics for converging vs. non-converging entities.

convergence of the learning process. Surprisingly, we observed that if there is one group of entities for which the magnitude of the updates stabilizes to very low values, indicating convergence, there is also a second group, which keeps receiving significant updates, with the repeating trend of random peaks indicating a failed convergence, rather than an insufficient amount of training (see Figure 3a).

Digging deeper into the characteristics of the two groups, we found a significant difference in the degree distributions, with the convergent group having a much higher degree (Figure 3b). Nonetheless, it would still contain entities with degree close to those of the non-converging group and so, to reduce the confounding factors of the analysis, we focused on this subset.

Comparing it to non-converging entities, we found that a higher degree imbalance in the *training* triples favors convergence (Figure 3c). Paired with the observed difficulty of predicting degree imbalanced triples at test-time, this tells us that the convergence of low-degree nodes is just a proxy for a strong overfitting. This is the case for `J-Pop` in our example above: it converges as only one relation type characterizes its neighborhood, but that also leads to overfitting the embedding. On the contrary, low-degree nodes connected to lower degree entities keep moving around the space, posing an equally hard challenge for the model at test-time in light of their instability.

Therefore, the challenge underlying the most imbalanced triples in the test set is two-folded: a strong overfitting on the one hand and an extremely poor learning on the other. We set to address this issue by proposing IMBALANCE, an inference-time latent search method that tries to improve model prediction on the most imbalanced triples involving low-degree entities.

## 4 INFERENCE TIME LATENT SEARCH FOR DEGREE IMBALANCE

### 4.1 PRELIMINARY AND NOTATION

A Knowledge Graph $\mathcal{G} = \{(s, p, o)\} \subseteq \mathcal{E} \times \mathcal{R} \times \mathcal{E}$ is a set of triples $t = (s, p, o)$, each including a subject (*head*) $s \in \mathcal{E}$, a predicate $p \in \mathcal{R}$, and an object (*tail*) $o \in \mathcal{E}$, where $\mathcal{E}$ and $\mathcal{R}$ are the sets of all entities and relation types, respectively. We refer to the task of predicting unseen triples in a KG as *Link Prediction*. It is formalized in literature as a learning-to-rank problem, where the objective

is learning a scoring function $f : \mathcal{E} \times \mathcal{R} \times \mathcal{E} \to \mathbb{R}$ that, given an input triple $t = (s, p, o)$, assigns a score $f(t) = f((s, p, o)) \in \mathbb{R}$ proportional to the likelihood that the fact $t$ is true.

## 4.2 INTUITION

At the end of the training of a conventional KGE model, every entity and relation has a low-dimensional, continuous representation which is the result of updates based on all triples in the KG. As such, it is the representation of the entity that best fits *all* training triples. Given a query $q$, however, a one-size-fits-all representation of the anchor might not be precise enough to find the correct answer, especially if the anchor has high degree, which could thus introduce a lot of noise into the representation. Consider again the example above and the node `Jackie Chan`. All the relations in its neighborhood are different from `/music/genre/artists` except for one. As a result, the embedding will have little pertinence with the query. This is the reason why we refine its embedding with the training triples sharing the anchor and the predicate of $q$, i.e., (`Mandopop,` `/music/genre/artists, Jackie Chan`). In this way, the new representation is the one that best suits $q$ and not *all* triples in the KG. We call this set the *training context* of $q$ and denote it $\mathcal{C}_q = \{(s, p, o_i) | o_i \in \mathcal{E}\} \subset \mathcal{G}$ if $q = (s, p, ?)$ and, analogously, $\mathcal{C}_q = \{(s_i, p, o) | s_i \in \mathcal{E})\} \subset \mathcal{G})$ if $q = (?, p, o)$.

The second issue of the optimization process we described above is related to low-degree entities. As we saw in the previous section, their embeddings are either heavily dependent on the few entities they are connected to and thus very prone to overfitting, or they suffer from an incomplete learning. This makes their reconstruction from the anchor node $s$ very unlikely, as their representation share too little with $s$. The node `J-Pop`, for example, is only connected to musicians, while Jackie Chan is best known for his acting. Similarly, other possible genres that are plausible answers for the query could be far off from Jackie Chan for the same reason. Therefore, it is essential to bias their embeddings in favor of the query. To do so, we leverage an external oracle that has the advantage of being degree agnostic, to limit the risk of neglecting low-degree entities. This oracle suggests triples similar to those in the training context of the query. In this way, we will bring the embeddings of Jackie Chan and musical genres similar to `Mandopop` closer together, making a correct answer to the query more likely.

## 4.3 IMBALANCE

Inference-time latent search has been successfully applied to improve generalization and model predictions in a completely different domain by Bonnet & Macfarlane (2024). As the degree imbalance could benefit from these properties, we design an inference-time latent search that can be selectively applied to target queries (see Algorithm 1). Built on top of a pre-trained KGE model with scoring function $f$, it takes a query $q = (s, p, ?)$ (analogous for $(?, p, o)$) and refines the representation of relevant entities optimizing for $n$ epochs a two-term objective function:

$$\mathcal{L}(q = (s, p, ?)) = \sum_{t^+ \in \mathcal{C}_q} f(t^+) + \sum_{t \in \Omega_q} f(t).$$

Crucially, our approach does not require any negatives, thus circumventing the problem of their synthetic generation that has attracted a lot of attention for its multiple criticalities (Kamigaito & Hayashi, 2022; Madushanka & Ichise, 2024). This aspect also limits the computational overhead of this method, that is extremely scalable. Moreover, it is extremely flexible, as it can be selectively applied to single queries, thus allowing to refine the prediction on single triples of interest.

**Training Context Term** The first term of the loss function sums the scores assigned to the training context of the query. These are ground truth triples that were already processed during training, and that the model should have already learned. However, as we observed above, the final embedding assigned to the anchor node is a one-size-fits-all representation that could under perform on a specific query. Therefore, during our latent search, we present these relevant triples again, to better tailor the anchor representation for the final prediction. Importantly, in this term we only optimize the embedding of the anchor node of the query, while keeping the others frozen. If we were to optimize the embeddings of head, tail and predicate, we would overfit learned triples even more, thus worsening the already poor generalization of the model at test-time. Limiting the optimization to the

---

**Algorithm 1** Inference-Time Latent Search for Degree Imbalance

---

**Require:** Pretrained KGE model with scoring function $f$ and entity embedding matrix $\mathbf{E}$, a query $q = (s, p, ?)$, training context $\mathcal{C}_q$ and oracle triples $\Omega_q$, number of latent search iterations $T$
**Ensure:** Ranked list of entities
 1: $\mathbf{E}_0 \leftarrow \mathbf{E}$
 2: Set the embeddings of $s$ to $\mathbf{s}^0 \leftarrow \mathbf{E}_0[s]$
 3: Set the embeddings of $o_j \in \{o | (s, p, o) \in \Omega_q\}$ to $\mathbf{o}_j^0 \leftarrow \mathbf{E}_0[o_j], \forall j = 1, ..., |\Omega_q|$
 4: **for** iteration $= 0, \ldots, T$ **do**
 5:     Compute $\mathcal{L}(q)$ and gradients w.r.t. $\mathbf{s}^i$ and $\mathbf{o}_j^i, j = 1, ..., |\Omega_q|$
 6:     Update $\mathbf{s}^{i+1} \leftarrow \text{GRADIENTUPDATE}(\mathbf{s}^i)$
 7:     Update $\mathbf{o}_j^{i+1} \leftarrow \text{GRADIENTUPDATE}(\mathbf{o}_j^i), \forall j = 1, ..., |\Omega_q|$
 8:     Update $\mathbf{E}_{i+1}[s] \leftarrow \mathbf{s}^{i+1}$ and $\mathbf{E}_{i+1}[o_j] \leftarrow \mathbf{o}_j^{i+1}, \forall j = 1, ..., |\Omega_q|$
 9: **end for**
10: Evaluate $q$ with the updated embeddings and extract ranked list of candidates
11: **return** Ranked candidates based on final scores

---

anchor node, on the contrary, we refine its representation to best suit the training context, improving its generalization and ring-fencing it in an area of the embedding space suitable for the query.

**Oracle-Enhanced Term** If the first term improves the representation of the anchor, we might still get bad predictions if the embeddings of the nodes answering the query are inaccurate. This is particularly likely for low degree entities that are dependent on very few facts that could well be irrelevant for the target query $q$, or, as shown before, prone to overfitting or susceptible to a failed learning. Therefore, refining the anchor node only using the training context is not enough. We need to adjust the target embeddings, biasing them toward the query. Doing so is not trivial, as, at test-time, we have no sense of what the correct answers to the query are. Moreover, being our focus on low-degree entities, little information about them is available within the KG. For this reason, we enhance our latent search with a set of additional triples $\Omega_q = \{(s, p, o_i) | o_i \in \mathcal{E}\}$ generated by an oracle that considers them likely answers to $q$. This time around, we optimize both the anchor and the target entity embeddings, so that, following the oracle leads, also the landscape of the answer representations changes in favor of the query.

As we said, the KG provides limited information about the target entity. Therefore, our oracle can be any out-of-band source of information well aligned with the task at hand. For our experiments on benchmark datasets we have used a Large Language Model (LLM) defining similarity on the encoded textual descriptions of the entities (see Appendix C for the technical details). In a different scenario, like the gene-disease association example, the oracle can be formalized as the preference of biologists for a subset of genes involved in a biological pathway relevant for the disease of interest or any other source of expertise. A final observation is needed here: the oracle triples may include false positives, that could surface up in the evaluation ranking. However, this risk is mitigated by the regularization effect it has on the representation of the anchor and by biasing the representation of possible targets toward the query.

## 5 EXPERIMENTS

### 5.1 EXPERIMENTAL SETTINGS

**Datasets and Test Splits.** We evaluate IMBALANCE on three encyclopedic benchmark KGs, widely used in literature: FB15k-237 (Toutanova & Chen, 2015), WN18RR (Dettmers et al., 2018) and YAGO3-10 (Mahdisoltani et al., 2015). To prove its efficacy on highly imbalanced triples involving *low-degree* entities, we identify *low-degree* nodes as those with degree below the first quartile of the degree distribution, and *high-degree* nodes as those with degree greater than the third quartile. The statistics of the resulting splits are reported in Table 8, while additional statistics about the dataset are reported in Appendix B.

**Evaluation Protocol and Metrics.** We evaluate IMBALANCE by corrupting the low-degree entity of the test triples with all entities in the KG and we consider the *filtered* setting (Bordes et al., 2013), i.e., we filter from the corruptions all facts in the training, validation or test sets. We then rank test

**Table 2:** Statistics of the datasets.

| Dataset | #H Nodes (Min degree) | #L Nodes (Max degree) | #Valid H-L | #Valid L-H | #Test H-L | #Test L-H |
|---------|----------------------|----------------------|-----------|-----------|----------|----------|
| FB15k-237 | 3536 (41) | 4015 (11) | 338 | 775 | 396 | 942 |
| WN18RR | 2296 (5) | 32 697 (10) | 295 | 689 | 277 | 753 |
| Yago3-10 | 28 913 (5) | 31 487 (16) | 25 | 298 | 35 | 270 |

**Table 3:** Results on the three benchmark datasets on the High-Low and Low-High triples splits. The best value for each metric on each dataset is reported in **bold** except in case of a tie, when they are underlined.

| Dataset | Model | High-Low | | | | Low-High | | | |
|---------|-------|------|------|------|-------|------|------|------|-------|
| | | MRR | H@1 | H@3 | H@10 | MRR | H@1 | H@3 | H@10 |
| FB15k-237 | ComplEx-N3 | 0.03 | 0.01 | 0.01 | 0.06 | 0.03 | 0.01 | 0.02 | 0.08 |
| | +IMBALANCE | 0.13 | 0.05 | **0.14** | 0.28 | 0.08 | 0.04 | 0.08 | 0.17 |
| | RotatE | 0.02 | 0.01 | 0.01 | 0.05 | 0.04 | 0.01 | 0.04 | 0.09 |
| | +IMBALANCE | 0.13 | **0.06** | 0.11 | **0.36** | 0.08 | 0.04 | **0.09** | 0.17 |
| Yago3-10 | ComplEx-N3 | 0.06 | 0.00 | 0.03 | 0.23 | 0.07 | 0.02 | 0.07 | 0.17 |
| | +IMBALANCE | 0.06 | 0.00 | 0.03 | 0.23 | **0.09** | 0.02 | **0.11** | **0.21** |
| | RotatE | 0.18 | 0.14 | 0.20 | 0.26 | 0.03 | 0.01 | 0.03 | 0.05 |
| | +IMBALANCE | **0.19** | 0.14 | 0.20 | **0.29** | 0.03 | 0.01 | 0.03 | 0.08 |
| WN18RR | ComplEx-N3 | 0.46 | 0.40 | 0.47 | 0.55 | 0.28 | 0.23 | 0.28 | 0.38 |
| | +IMBALANCE | 0.48 | 0.40 | 0.51 | **0.61** | 0.31 | 0.25 | 0.32 | **0.46** |
| | RotatE | 0.49 | 0.45 | 0.50 | 0.57 | 0.32 | 0.27 | 0.33 | 0.42 |
| | +IMBALANCE | **0.51** | **0.46** | **0.55** | 0.60 | **0.33** | 0.27 | **0.35** | 0.44 |

triples against all corruptions. The metrics used are the usual for link prediction: Mean Reciprocal Rank (MRR) and Hits at N (Hits@N).

**Hyperparameter-Search** Given a pre-trained KGE model, IMBALANCE has only three hyper-parameters: the learning rate $\lambda$ of an Adam (Kingma & Ba, 2015) optimizer, the number of latent search iterations and the number $m$ of oracle-generated triples included in $\Omega_q$. We selected optimal values based on the validation performance using MRR as the reference metric.

## 5.2 RESULTS

**IMBALANCE Impact** The results of our experiments are reported in Table 3. The impact of the latent search on FB15k-237 triples is striking, with metrics that improved at least by a factor of 2 on the Low-High split and over a factor of 6 on the High-Low one. This shows how IMBALANCE compensates the flaws of the pre-trained models. Also on WN18RR there is a noticeable improvement across all metrics, but it is smaller than on FB15k-237. The reason for it is that in WN18RR the degree distribution is concentrated around values much smaller than those of FB15k-237 (see Table 8). This reduced polarization results in a less pressing degree imbalance issue, making the improvement of IMBALANCE less striking. Finally, on Yago3-10, we only see an improvement on the Hits@10. This behavior can be justified by the fact that the oracle generated the triples for this dataset using only the labels of the entities of the KG, while it leveraged labels and additional descriptions for FB15k-237 and WN18RR. This impacted negatively the similarity of triples extracted by the oracle, introducing less relevant triples in the latent search that prevented the correct answers to reach the top positions of the ranking.

**Baseline Comparison** We compare the impact of our method with two different approaches in Table 4. KG-Mixup (Shomer et al., 2023) was designed to counter a vague notion of degree bias and to enhance performance on low-degree entities. However, we show how it provides little benefit on imbalanced triples compared to IMBALANCE. CSProm-KG (Chen et al., 2023), on the other hand, is a method that integrates an LLM to enhance link prediction and is way superior in aggregate metrics to ComplEx and RotatE ($MRR = 0.36$, $H@10 = 0.54$ on the entire FB15k-237 test set). On imbalanced triples, however, IMBALANCE outperforms it or falls behind by a slight margin. Therefore, our approach makes simpler models perform on the same level as more involved ones

**Table 4:** Results on FB15k-237 for IMBALANCE and two baselines. The best value for each metric on each dataset is reported in **bold** except in case of a tie, when they are underlined.

| Dataset | Model | High-Low | | | | Low-High | | | |
|---|---|---|---|---|---|---|---|---|---|
| | | MRR | H@1 | H@3 | H@10 | MRR | H@1 | H@3 | H@10 |
| FB15k-237 | **ComplEx-N3 + IMBALANCE** | 0.13 | 0.05 | **0.14** | 0.28 | 0.08 | 0.04 | 0.08 | 0.17 |
| | **RotatE + IMBALANCE** | 0.13 | **0.06** | 0.11 | **0.36** | 0.08 | 0.04 | 0.09 | 0.17 |
| | **TuckER** balazevic2019tucker | 0.08 | 0.03 | 0.08 | 0.21 | 0.08 | 0.04 | 0.07 | 0.16 |
| | **+KG-Mixup** shomer2023degree$_b ias$ | 0.09 | 0.03 | 0.08 | 0.24 | 0.08 | 0.03 | 0.08 | 0.17 |
| | **CSProm-KG** chen2023cspromkg | 0.09 | 0.02 | 0.07 | 0.29 | **0.09** | **0.05** | 0.09 | 0.17 |

**Table 5:** Results on FB15k-237 isolating the contribution of the two terms of the loss function.

| Dataset | Model | High-Low | | | | Low-High | | | |
|---|---|---|---|---|---|---|---|---|---|
| | | MRR | H@1 | H@3 | H@10 | MRR | H@1 | H@3 | H@10 |
| FB15k-237 | **ComplEx-N3** | 0.03 | 0.01 | 0.01 | 0.06 | 0.03 | 0.01 | 0.02 | 0.08 |
| | **+Context Only** | 0.05 | 0.01 | 0.04 | 0.14 | 0.05 | 0.02 | 0.04 | 0.12 |
| | **+Oracle Only** | 0.10 | 0.05 | 0.09 | 0.22 | 0.08 | 0.03 | 0.07 | 0.17 |
| | **+IMBALANCE** | 0.13 | 0.05 | **0.14** | 0.28 | 0.08 | 0.04 | 0.08 | 0.17 |
| | **RotatE** | 0.02 | 0.01 | 0.01 | 0.05 | 0.04 | 0.01 | 0.04 | 0.09 |
| | **+Context Only** | 0.06 | 0.02 | 0.06 | 0.16 | 0.06 | 0.02 | 0.06 | 0.14 |
| | **+Oracle Only** | 0.12 | 0.05 | 0.10 | 0.34 | 0.08 | 0.04 | 0.09 | 0.17 |
| | **+IMBALANCE** | 0.13 | **0.06** | 0.11 | **0.36** | 0.08 | 0.04 | 0.09 | 0.17 |

and does so preserving the computational efficiency that CSProm-KG and other LLM-enhanced KGE models cannot offer.

**Loss function terms contribution** To gauge the contribution of the training context term and of the oracle term, we run separate experiments where we switch off alternately one or the other. The results for FB15k-237 are reported in Table 5, while those for WN18RR and Yago3-10 are in Appendix D. Numbers clearly show that both terms provide a positive contribution on their own. This supports both our claims on how grounding the representation of the anchor node in a query-friendly way is essential and on the importance of adjusting the embeddings of other entities. We explain the bigger impact of the oracle term as it also contributes directly to the optimization of the anchor embedding. Finally, the joint contribution of the two outperforms the single terms, supporting their complementarity.

**Time-Complexity and Latency** The time-complexity needed to run one epoch of IMBALANCE is $\mathcal{O}\left(k \cdot (|\mathcal{C}_q| + |\Omega_q|)\right)$. In fact, it applies the KGE scoring function and computes the gradients for the embeddings of the training-context and oracle-generated triples. The already low time-complexity to train a traditional KGE training for one epoch amounts to $\mathcal{O}(|\mathcal{G}|(\eta+1)k)$, where $\mathcal{G}$ is the training graph and $\eta$ the number of negatives. Instead of re-traning such model, applying IMBALANCE to a test set $\mathcal{T}$ is more efficient, as the size of $\mathcal{T}$ is typically orders of magnitude smaller than $\mathcal{G}$. Moreover, $\mathcal{C}_q$ is a limited subset of $\mathcal{G}$ and the number of oracle triples are only a few dozens at most (in our experiments always $\leq 50$), yielding $|\mathcal{T}| \cdot (|\mathcal{C}| + |\Omega|) \ll N$. Finally, IMBALANCE requires fewer epochs to reach convergence, thus confirming its efficiency even compared to the strong baseline. The run time latency of the system is reported in Table 6.

## 6 LIMITATIONS AND FUTURE DIRECTIONS

If this work sheds a light on a key issue affecting KGE models, it still has limitations that we will address in future work. First, IMBALANCE can only be applied on queries for which the training context is non-empty. Overcoming this limitation would mean refining the selection of training triples *tightly related* to the query. This would be of great interest, as it would explain which triples have the highest impact on a prediction. Second, we proved its applicability to High-Low and Low-High triples, but it remains a challenge to understand how it could benefit triples where the imbalance is more limited. The issue is limited as IMBALANCE can be applied selectively on single

**Table 6:** Latency of IMBALANCE compare to re-training the KGE model from scratch. The runtime is reported in seconds over 30 epochs of latent search for IMBALANCE, though we typically observe convergence within the first 10 epochs.

|  | Model | FB15k-237 | Yago3-10 |
|---|---|---|---|
| Single query | IMBALANCE | 0.91 | 1.39 |
| Full test set | IMBALANCE | 440.7 | 400.1 |
| Pre-training | RotatE | 935.9 | 24,131.3 |

triples, but an extension that benefits all triples could stretch the applicability to wider use cases. Finally, the quality of the triples generated by the oracles affects the quality of the results, which made our results dependent on the LLM. We leave for future work an in-depth study of different oracles, that go beyond LLMs and potentially include human feedback.

## 7 CONCLUSIONS

This work has introduced the degree imbalance problem, an issue that heavily affects a wide variety of KGE models and hinders their predictive power. We provided deep insights on the learning issue from which the problem stems, proving how low-degree entities are impacted by strong overfitting or by a failed convergence. To mitigate the issue, we proposed IMBALANCE, the first inference-time latent search method applied in the realm of KGE embeddings. We validated its efficacy by obtaining reliable predictions on highly imbalanced triples, preserving model efficiency, and opening the door to the application of this method in use cases where degree imbalance is a consistent concern.

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

## A  DEGREE IMBALANCE FOR ADDITIONAL MODELS

In Table 7 we report the aggregate metrics for TransE and DistMult.

In Figure 4 we report the degree imbalance for TransE and DistMult. As we can see, the behaviour resembles that of ComplEx and DistMult.

**Table 7:** Aggregate performance of KGE models on FB15k-237 and YAGO3-10

|          | FB15k-237 |       | YAGO3-10 |       |
|----------|-----------|-------|----------|-------|
|          | **MRR**   | **H@10** | **MRR** | **H@10** |
| TransE   | 0.31      | 0.49  | 0.35     | 0.55  |
| DistMult | 0.30      | 0.48  | 0.34     | 0.53  |

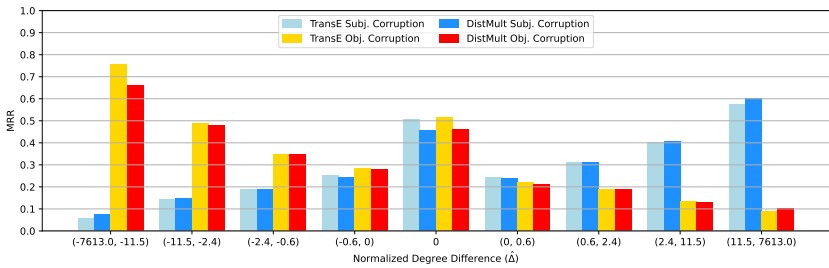

**(a)** FB15k-237

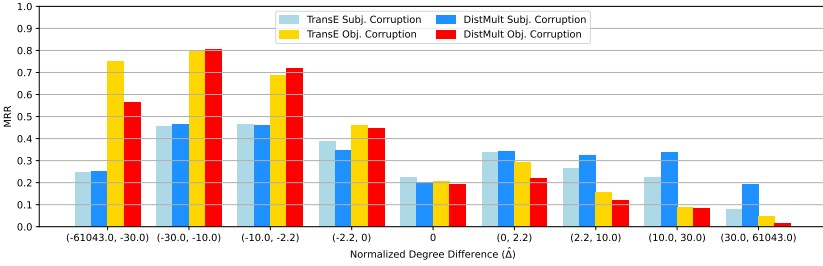

**(b)** YAGO3-10

**Figure 4:** Performance of conventional KGE models on test triples binned by normalized degree difference. Bins are obtained using quartiles of $|\hat{\Delta}(t)|$ for $t \in \mathcal{G}$. On the left hand side we have triples with high-degree object and low-degree subject, on the right hand side, high-degree subjects and low-degree objects. We isolate subject and object corruption.

As we can see from the Figure 5, NBFNet (Zhu et al., 2021), despite its superior performance, suffers from the same degree imbalance issue as conventional KGE models. Compared to the results in Figure 2, the individual performance on most buckets is slightly better.

## B  DATASET STATISTICS

Statistics of the three datasets are reported in Table 8.

**Table 8:** Statistics of the datasets.

| Dataset   | #Entities | #Relations | #Train    | #Valid | #Test  |
|-----------|-----------|------------|-----------|--------|--------|
| FB15k-237 | 15 541    | 237        | 272 115   | 17 535 | 20 466 |
| WN18RR    | 40 943    | 18         | 86 835    | 3034   | 3134   |
| Yago3-10  | 123 182   | 37         | 1 079 040 | 5000   | 5000   |

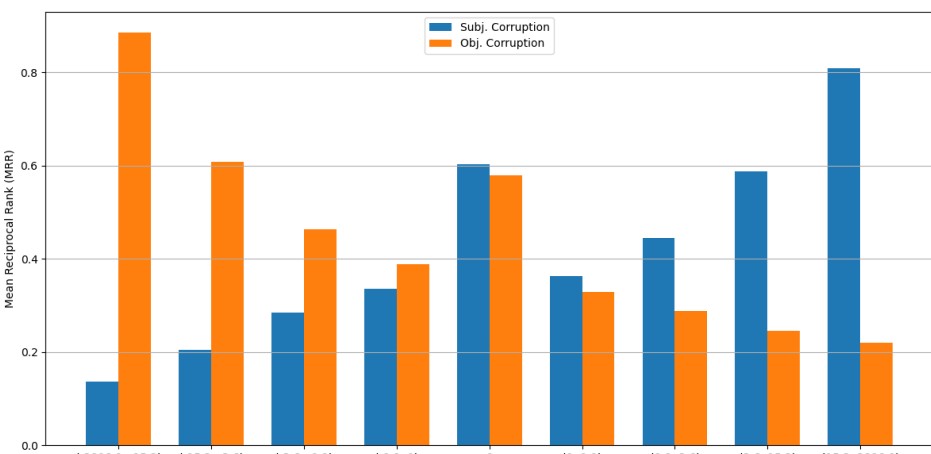

**Figure 5:** Performance of NBFNet on test triples binned by normalized degree difference. Bins are obtained using quartiles of $|\hat{\Delta}(t)|$ for $t \in \mathcal{G}$. On the left hand side we have triples with high-degree object and low-degree subject, on the right hand side, high-degree subjects and low-degree objects. We isolate subject and object corruption.

## C  ORACLE TECHNICAL DETAILS

For all our experiments, we have used `NovaSearch/stella_en_400M_v5`[1] with an embedding dimension of 1,024 to encode the labels and descriptions of the entities in our datasets (with the exception of Yago3-10, for which we used only the labels). Given $q = (s, p, ?)$, to generate the $m$ triples in $\Omega_q$, we have extracted the top-$m$ entities closest to the barycenter of the set $\{e_j \in \mathcal{G} | \{(s, p, e_j) \in \mathcal{C}_q\}$ in the embedding space of the LLM. The distance between entities was computed using the cosine similarity.

As querying an LLM can introduce a significant overhead, we have extracted the LLM-triples as a preprocessing step ahead of running the experiments. In this way, the method remains lightweight.

## D  ADDITIONAL EXPERIMENTS

**Loss Function Terms Contribution**   See results in Table 9. As the improvement in Table 3 showed a smaller increase on these datasets compared to FB15k-237, it is harder to appreciate significant differences in the contribution of the separate terms of the loss. However, the aggregation of the two yields the best results.

**Sankey Plots**   To further explore how IMBALANCE impacts the ranks assigned to test triples, we leverage Sankey plots that show how ranks flow from the values assigned by the pre-trained model (on the left), to the ranks assigned using IMBALANCE (on the right). From Figures 8-9-10 we can observe how IMBALANCE improves ranks of hundreds of positions. Despite the metrics observed in Table 3 do not necessarily show it, bringing up ranks that are $> 500$ at the top positions of the ranking is a huge achievement, testifying the value of the approach.

**Degenerate Behaviour Loss Function**   The loss function is purely maximization over a set of known positive facts ($\mathcal{C}_q$) and oracle-suggested positive facts ($\Omega_q$). This might lead to questioning the stability of the optimization process, with the risk of collapse or severe overfitting. We did not observe such behaviour, as the design of ImbalancE prevents it for two reasons:

- The first term of the loss function is used to optimize only the embedding of the anchor node, while the embeddings of the target entities of the training context remain fixed, thus preventing a collapse of the embeddings.

---

[1]`https://huggingface.co/NovaSearch/stella_en_400M_v5`

**Table 9:** Results on WN18RR and YAGO3-10 isolating the contribution of the two terms of the loss function.

| Dataset | Model | High-Low | | | | Low-High | | | |
|---|---|---|---|---|---|---|---|---|---|
| | | MRR | H@1 | H@3 | H@10 | MRR | H@1 | H@3 | H@10 |
| Yago3-10 | ComplEx-N3 | 0.06 | 0.00 | 0.03 | 0.23 | 0.07 | 0.02 | 0.07 | 0.17 |
| | +Context Only | 0.06 | 0.00 | 0.03 | 0.23 | 0.06 | 0.02 | 0.07 | 0.17 |
| | +Oracle Only | 0.06 | 0.00 | 0.03 | 0.23 | 0.07 | 0.00 | 0.10 | 0.20 |
| | +IMBALANCE | 0.06 | 0.00 | 0.03 | 0.23 | **0.08** | 0.02 | **0.11** | **0.20** |
| | RotatE | 0.18 | 0.14 | 0.20 | 0.26 | 0.03 | 0.01 | 0.03 | 0.05 |
| | +Context Only | 0.18 | 0.14 | 0.20 | 0.26 | 0.04 | 0.02 | 0.03 | 0.08 |
| | +Oracle Only | 0.19 | 0.14 | 0.20 | 0.29 | 0.03 | 0.01 | 0.03 | 0.06 |
| | +IMBALANCE | **0.19** | 0.14 | 0.20 | **0.29** | 0.03 | 0.01 | 0.03 | 0.08 |
| WN18RR | ComplEx-N3 | 0.46 | 0.40 | 0.47 | 0.55 | 0.28 | 0.23 | 0.28 | 0.38 |
| | +Context Only | 0.45 | 0.40 | 0.47 | 0.55 | 0.28 | 0.22 | 0.28 | 0.38 |
| | +Oracle Only | 0.48 | 0.40 | 0.51 | 0.61 | 0.31 | 0.25 | 0.33 | 0.45 |
| | +IMBALANCE | 0.48 | 0.40 | 0.51 | 0.61 | 0.31 | 0.25 | 0.32 | **0.46** |
| | RotatE | 0.49 | 0.45 | 0.50 | 0.57 | 0.32 | 0.27 | 0.33 | 0.42 |
| | +Context Only | 0.49 | 0.45 | 0.5 | 0.58 | 0.32 | 0.27 | 0.33 | 0.42 |
| | +Oracle Only | 0.49 | 0.43 | 0.53 | 0.6 | 0.32 | 0.27 | 0.34 | 0.42 |
| | +IMBALANCE | **0.51** | **0.46** | **0.55** | 0.60 | **0.33** | 0.27 | **0.35** | 0.44 |

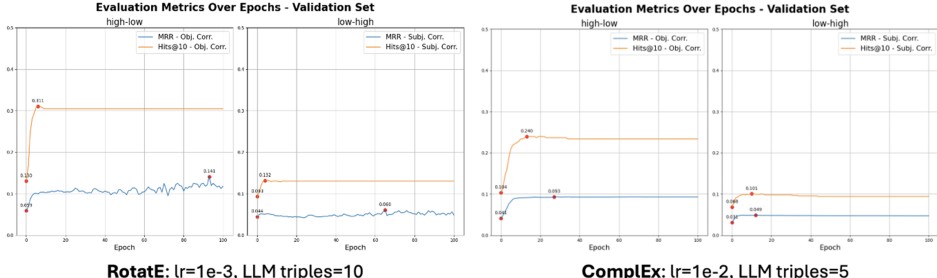

**Figure 6:** Validation metrics for FB15k-237 extending the latent search to 100 epochs.

- Similarly, in both terms of the loss function, the embedding of the relation is kept constant, providing a similar constraint that prevents an excess of overfitting.

To further validate such a claim, we ran ImbalancE on top of our models for 100 epochs for FB15k-237. As evident from Figure 6, MRR and H@10 on the validation set plateaus after the first few epochs and then remain pretty much constant, showing non-degenerate behavior. Moreover, we remark once more how ImbalancE is meant to be used at inference time: as such, we expect to run it for a limited number of iterations, as plots below show is recommendable.

**Training Context Size v Performance** We try to understand whether there is any correlation between the size of the training context and the improvement that IMBALANCE achieves on imbalanced triples. However, Figure 7 shows how there is no such correlation, showing that the method can benefit imbalanced triples independently on the amount of information available in the training data, proving once more that the true issue lies in the degree imbalance.

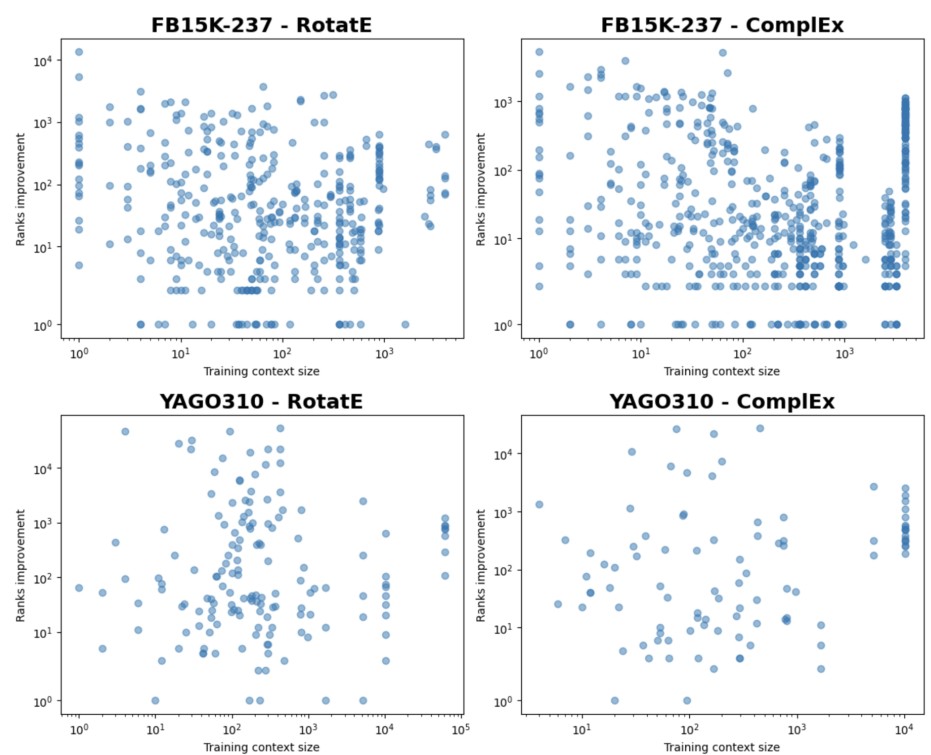

**Figure 7:** Correlation between the size of the training context and the rank improvement in FB15k-237 and Yago3-10.

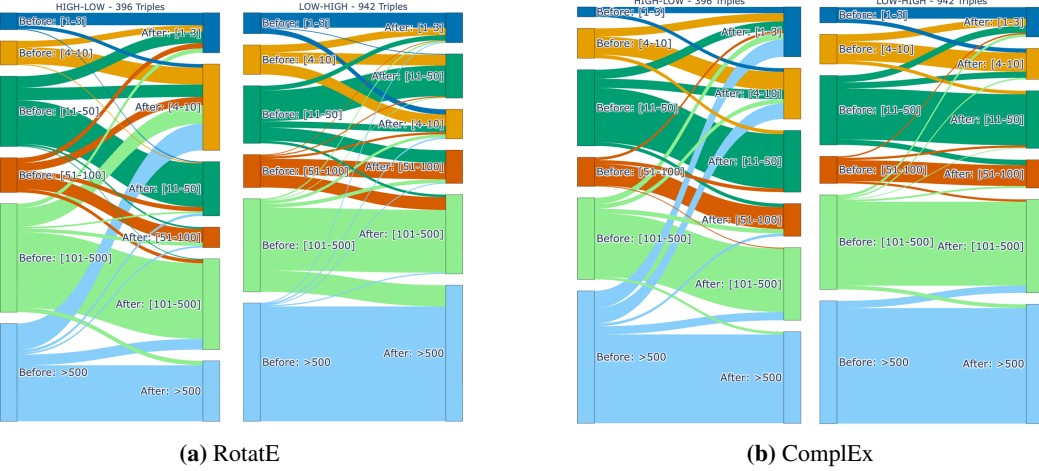

(a) RotatE               (b) ComplEx

**Figure 8:** Sankey plots detailing how IMBALANCE altered RotatE and ComplEx ranks on FB15k-237.

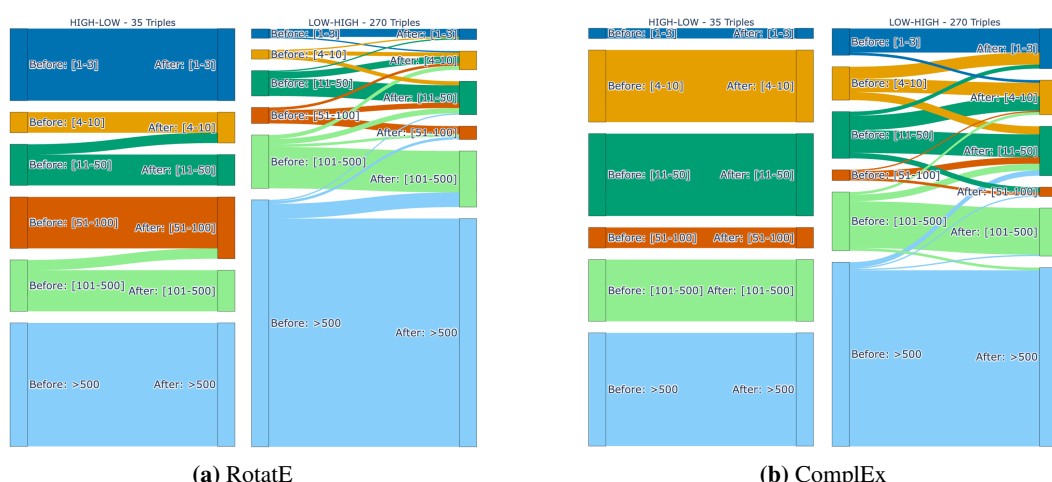

(a) RotatE

(b) ComplEx

**Figure 9:** Sankey plots detailing how IMBALANCE altered RotatE and ComplEx ranks on Yago3-10.

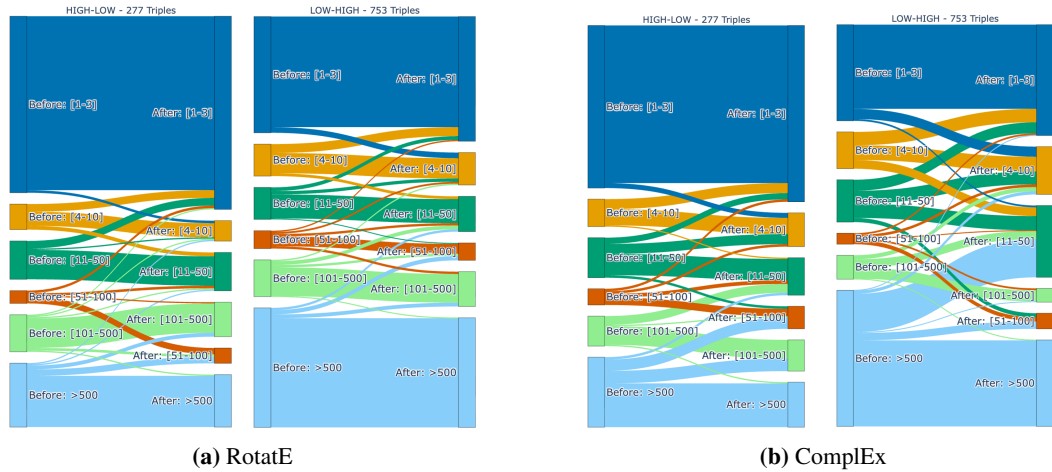

(a) RotatE

(b) ComplEx

**Figure 10:** Sankey plots detailing how IMBALANCE altered RotatE and ComplEx ranks on WN18RR.

