# OpenReview forum: "ImbalancE: Inference-Time Latent Search against Degree Imbalance in Link Prediction"
_ICLR.cc/2026/Conference — ICLR 2026 Conference Withdrawn Submission_

### Official Review · Reviewer_Hjxb · 2025-10-22

**Soundness:** 3
**Presentation:** 2
**Contribution:** 3
**Rating:** 6
**Confidence:** 3

**Summary:**

This paper identifies degree imbalance as a key weakness in Knowledge Graph Embedding (KGE) models—low-degree entities are poorly predicted when paired with high-degree ones. To fix this, the authors propose IMBALANCE, an inference-time latent search that refines entity embeddings using query-specific context and oracle-generated triples. Experiments on benchmark datasets show that IMBALANCE significantly improves link prediction for imbalanced triples.

**Strengths:**

- This paper addresses Knowledge Graph Completion (KGC) under conditions of large degree differences between entities, which is a novel and practically valuable problem.
- This paper introduces IMBALANCE, a practical inference-time optimization method that enhances pretrained models without requiring retraining.
- The proposed method can be applied to existing KGE models and demonstrates good scalability.
- Extensive experiments on benchmark datasets show significant and consistent improvements, proving the method’s effectiveness.

**Weaknesses:**

- The datasets evaluated in this paper are all standard benchmark KGC datasets, while the introduction mentions practical scenarios such as drug–target discovery and recommender systems. It would be valuable to further investigate the method’s performance on knowledge graphs from these real-world applications.
- Since the proposed approach introduces external knowledge through an oracle or LLM-based augmentation, the comparison with baselines may not be entirely fair. It is recommended to include baseline models that also leverage external knowledge for a more equitable evaluation.
- The training-context term only incorporates related (s,p,?) triples. How does the method handle potential noise in the knowledge graph? How is the importance of these triples determined?
- Apart from the FB15k‑237 dataset, the performance gains on the other two datasets are relatively small. Additional ablation studies on these datasets would help strengthen the experimental evidence.
- The number of collected test samples seems limited.
- It is suggested to express the results in percentage form to clarify the proportion of improved cases.

**Questions:**

Please refer to the Weaknesses.

---

> ### Author Response · Authors · 2025-12-04
> **Review answer**
>
> Thanks for the positive evaluation of your work. It is great that you appreciated the relevance and impact of the degree imbalance problem, and appreciated ImbalancE in its key features: being a plug-in approach, scalable and effective model.
>
> Weakness 1: unfortunately, the short timeline did not allow us to run experiments on new benchmarks, mainly because this would require gathering descriptions for the entities in these datasets. We aimed at doing it on the biomedical knowledge graph, which would be particularly pertinent. However, the information gathered into KGs in this domain is scattered across multiple data sources, that often present various accessibility challenges. We leave this extension for future work.
>
> Weakness 2: we have added additional baselines in the paper as per your request.
>
> Weakness 3: as per the noise in the KG, ImbalancE is as vulnerable to it as the pretrained model is. Data cleansing over the training data would help, however, the scale and nature of KG makes this step particularly challenging.
>
> Weakness 4: it is true that on some triples of Yago3-10 and WN18RR the gains are less marked compared to FB15k-237. We believe the reason for it is two folded: on one side, Yago3-10 and WN18RR have a degree distribution of the training set that is less skewed compared to FB15k-237. This makes the reach of the degree imbalance problem more limited, and hence also the contribution of ImbalancE is reduced. On the other side, ImbalancE seems to be particularly effective in improving the ranking overall, but sometimes struggles at nailing the very first positions of the rankings, which make the impact on H@1 and H@3 not noticeable. However, we have added in the appendix a series of Sankey plots that show how ImbalancE improves substantially (even of hundreds of positions) the ranks of triples. We hope these plots can show this benefit that cannot be observed with conventional metrics, but that is equally important especially in practical use cases.
>
> Weakness 5: we aim to investigate new benchmarks (like the ones mentioned in question 1) to show the impact of ImbalancE on a broader set of triples in future work.

---

### Official Review · Reviewer_Vq7R · 2025-10-25

**Soundness:** 3
**Presentation:** 3
**Contribution:** 2
**Rating:** 4
**Confidence:** 4

**Summary:**

The paper identifies degree imbalance (large head–tail degree gaps) as one of the key reason that the current KGE models underrank low-degree entities and their performance collapses as the normalized degree difference grows. They propose IMBALANCE, a test-time latent search plug-in: for a query q, run a few gradient steps only on the anchor and a few oracle-suggested candidate targets, optimizing a two-term positive-only loss 1) training-context triples for the anchor and 2) oracle triples. After that, they rank with the updated embeddings. The oracle uses text embeddings of entity labels/descriptions to retrieve plausible candidates (precomputed; cosine-based). On hard High↔Low subsets, IMBALANCE markedly boosts MRR/Hits@k across ComplEx/RotatE/TransE/DistMult (e.g., FB15k-237) and both loss terms matter in ablations. An qualitative example shows rank improving 67 → 3 after the latent search.

**Strengths:**

1.	This paper represents a new test-time adaptation method in the domain of KGE. Without any further training on all embeddings, they locally and temporally fine-tune the anchor embeddings for each test query.
2.	Clear diagnosis of the failure mode. They bin test triples by normalized degree difference and show performance drops as the gap grows (Figure 2), cleanly isolating the imbalance effect beyond relation-type heuristics. They track per-epoch embedding movement and uncover two low-degree failure modes-overfitting (premature convergence) and non-convergence (oscillation), which explain why low-degree answers are misranked. (Figure 3, analysis.)
3.	Performance gains on the hardest High-Low cases across multiple models/datasets. Ablations show both context and oracle are needed.
4.	Oracle triples are precomputed; the method is model-agnostic and lightweight compared to retraining.

**Weaknesses:**

1.	Oracle dependency. The improvements partially come from text (labels/descriptions). No comparison to text-augmented training baselines is provided, so the share of gain from the LLM side vs. the latent search isn’t isolated.
2.	MBALANCE's effectiveness depends on the choice and size of the training-context subset. The paper defines C_q (triples sharing the anchor and relation) but does not provide a principled selection policy or sensitivity analysis. For high-degree anchors, C_q can be large; different subsampling strategies (diversity-, hardness-, or degree-aware) may materially change both accuracy and latency. Please report results as |C_q| and the selection rule varies
3.	Even though no heavy re-training is needed for each inference, the proposed method still needs a few steps of gradient updates for each test query. This may cause significant latency compared to the other baselines.

**Questions:**

1.	Please specify the exact construction of the training-context set C_q: how many triples, how selected (by relation, recency, degree, similarity?), and whether selection differs by model/dataset. Your Algorithm 1 references C_q, but the selection criteria aren’t fully explicit.
2.	After the per-query latent search, do you re-rank over all entities or only over the oracle candidates? Algorithm 1 says "extract ranked list of candidates" but does not spell out the candidate set used for evaluation. Please clarify.
3.	You avoid negatives for efficiency. Did you observe any degenerate behaviors (e.g., indiscriminate score inflation) without a contrastive term, and how do you regularize against that? You note that Oracle triples may include false positives that can surface in ranking.
4.	How often did that occur in practice, and what guardrails (e.g., confidence thresholds, re-scoring) reduce harm? Any quantitative analysis?
5.	You acknowledge IMBALANCE requires a non-empty training context for the anchor. How does performance degrade as context shrinks, and can a "pure-oracle" fallback (no context) recover some of the gains?
6.	Since the oracle injects textual knowledge, please compare against text-aware KGE (e.g., adding description embeddings at training) to isolate the unique value of test-time latent search vs. simply having more information.
7.	Please provide per-query latency for a typical T, and how cost scales with |\Omega_q|=m and context size.

---

> ### Author Response · Authors · 2025-12-04
> **Review answer**
>
> Thanks for your insightful review and for appreciating the key contributions of our work.
>
> Weakness 1: we have added text-augmented baseline in the results section of the paper. Please refer to it.
>
> Weakness 2: ImbalancE selects the triples relevant to a query as all the training (and validation) triples that share the same (subject, predicate) (if q=(s, p, ?)) or (predicate, object) (if q = (?, p, o)) as those are the most directly related to the query. However, you are right, as stated in the limitations section, we don't exclude that having different ways to establish relevant triples for a query could lead to different training contexts that could further enhance the performance of the model. We aim to investigate such possibility in future work.
>
> Weakness 3: we have added a section on the time-complexity and latency of the method, that shows how lightweight it is. Sorry for not including it in the original version of the paper.
>
> Question 1: please refer to lines [284-286]
>
> Question 2: we re-rank against all candidates, in the conventional "filtered setting": for every target triple t = (s, p, o), we
> separately corrupt the subject and the object with all the entities in the KG not generating a fact in the
> training, validation or test sets. We then rank test triples against all corruptions.
>
> Question 3: your concern on the stability of the optimization is very reasonable, and one we also expected to observe. However, we experimentally found that no collapse happen. We have added a paragraph in appendix D (Degenerate Behaviour Loss Function).
>
> Question 4: see question 3.
>
> Question 5: we did not observe any correlation between the size of the training context and improvements. We have added a section in the appendix D to show it. A purely oracle-based fallback could actually be a possibility and, as the ablation study on the loss function terms shows, it could be an effective one.
>
> Question 6: added CSProm-KG as an additional text-aware KGE.
>
> Question 7: added dedicated section in the paper.

---

### Official Review · Reviewer_VZ92 · 2025-10-29

**Soundness:** 3
**Presentation:** 2
**Contribution:** 1
**Rating:** 2
**Confidence:** 4

**Summary:**

The paper identifies and analyzes the degree imbalance problem in Knowledge Graph Embeddings (KGEs)—the phenomenon where test triples with large degree differences between entities (e.g., high-degree head vs. low-degree tail) yield poor link prediction performance.
It introduces IMBALANCE, an inference-time latent search procedure that fine-tunes entity embeddings using a dual-term objective: (1) a training-context term to re-optimize the anchor entity, and (2) an oracle-enhanced term leveraging external LLM-derived triples to bias low-degree entities toward the query. Experiments on FB15k-237, WN18RR, and YAGO3-10 show substantial improvements in MRR and Hits@N for imbalanced triples, especially on FB15k-237.

**Strengths:**

1) The authors systematically connect prediction degradation to degree imbalance, offering solid empirical evidence (e.g., Figure 2, showing sharp MRR drop-offs by degree difference bins).

2) Simple yet general idea: The method can be plugged into any pretrained KGE without retraining, adding post-hoc refinement.

3) Dual-objective design: Separating the anchor refinement and oracle biasing terms is conceptually sound and empirically validated through ablations.

4) Application relevance: Addresses a realistic pain point for long-tail link prediction (e.g., drug discovery, recommender systems).

**Weaknesses:**

1) Limited novelty. The inference-time latent search idea is borrowed almost directly from prior optimization-based refinement methods (e.g., Bonnet & Macfarlane 2024). The contribution here is more contextual adaptation to KGEs than methodological innovation. The optimization objective is simplistic (a score-sum without any new regularization or causal insight)

2) Outdated backbone models. The work only tests on ComplEx-N3, and RotatE. State-of-the-art KGE methods in 2020-2025 such as NBFNet are absent, though I see that the method can be pluged into more models.

3) The authors evaluate IMBALANCE only as a plug-in to their own reimplementation, not against competing post-hoc correction or de-biasing methods (e.g., Shomer et al. 2023.Toward Degree Bias in Embedding-Based
Knowledge Graph Completion). Even simple baselines such as degree-aware normalization or dropout fine-tuning are missing.

4) Datasets are small and classical (FB15k-237, WN18RR, YAGO3-10). The improvements on YAGO3-10 are marginal (MRR unchanged, +0.02 H@10). The oracle fails on text-only datasets, suggesting brittleness. The gains largely come from low-baseline models (ComplEx/RotatE underperforming due to degree bias).

**Questions:**

1) How does IMBALANCE compare to other debiasing or reweighting methods (e.g., degree-regularized KGEs or post-hoc calibration)?

2) Why restrict to classical models? Could modern transformer-based KGE models also benefit?

3) How sensitive is the method to oracle noise? Have you tested with random oracle triples?

4) What is the runtime cost per query, and how does it scale to large KGs (e.g., Wikidata5M)?

5) Would a simple degree-normalized reweighting baseline achieve similar improvements?

---

> ### Author Response · Authors · 2025-12-04
> **Review answer**
>
> Thank you for your constructive feedback. Your comments helped us refine the documents with some key additions and refinement.
>
> Weakness 1: As it concerns the limited novelty of the idea, we believe it was not a trivial adaptation. In fact:
>
> •	the original work proposes an architecture that requires training beyond the latent search and requires approximating distributions, while our model doesn't.
>
> •	the original architecture is applied to a task very far from knowledge graphs, a type of data that poses additional challenges. For example, designing an inference-time latent search approach for KGE models requires disentangling as much as possible the intricate dependencies of different entities and relations.
>
> •	using the approach to tackle the degree imbalance issue is in itself a primary contribution, as nothing similar to it exists in the realm of the task solved in the original paper.
>
> Finally, we believe the simplicity of the scoring function contributes to keeping the underlying idea easy to grasp, stripping it away of complications that often provide very marginal gains.
>
> Weakness 2: as briefly mentioned in lines [139-141], NBFNet was not directly considered in the experimental section as its architecture uses a set of neural network weights to compute the representation of all entities in the context of a query. Therefore, it is impossible to disentangle the optimization of the anchor entity from the optimization of the other targets in the training context, which, as observed in lines [321-323] causes heavy overfitting.
>
> Weakness 3: we have added additional baselines to the experiment section, but we are not fully sure what baselines you hint at with "degree-aware normalization or dropout fine-tuning".
>
> Weakness 4: we picked classical datasets as they are typically those used as benchmarks by the community. We reckon they are small, however, ImbalancE is not dependent on the size of the underlying graph, that's why we sticked to smaller ones (though Yago3-10, already has a fair size). In relation to your comment on the gains coming from the low-baseline models, we would like to highlight that that's were the benefit of ImbalancE comes from: it can bridge the performance gap with more involved and complex models while remaining lightweight and scalable.
>
> Question 1: Could you please clarify what do you mean by debiasing or reweighting methods? Post-hoc calibration seems not to be pertinent, given the scope of our work.
>
> Question 2: we have added baselines in the experiment section of the paper.
>
> Question 3: the method seems to be pretty robust, as most of the oracle triples are not correct answer to a query.
>
> Question 4: we have added the time-complexity and latency to the paper.
>
> Question 5: could you please clarify?

---

### Official Review · Reviewer_YMSQ · 2025-10-29

**Soundness:** 3
**Presentation:** 3
**Contribution:** 2
**Rating:** 4
**Confidence:** 3

**Summary:**

The authors focused on the systematic performance degradation on triples involving entities with high degree imbalance. The proposal of an inference-time latent search method, IMBALANCE, to mitigate this bias is novel within the KGE domain. The empirical results showing significant gains on the hardest subset of test triples are compelling. However, the proposed objective function, combined with the core mechanism of gradient updates, introduces several critical theoretical and practical concerns that need to be addressed.

**Strengths:**

1. The paper provides a good analysis, defining the problem not merely as "degree bias," but specifically as degree imbalance between head and tail entities in test triples. The analysis linking this issue to two distinct learning failures (overfitting and failed convergence) in low-degree nodes is insightful.

2. The reported improvements on the highly imbalanced splits of FB15k-237 (metrics improving by factors up to x6) demonstrate the effectiveness on critical corner cases.

**Weaknesses:**

## Weakness 1

My primary theoretical concern lies in the structure of the dual-term objective function:

$$L(q = (s, p, ?)) = \sum_{t^+ \in \mathcal{C}_q} f(t^+) + \sum_{t \in \Omega_q} f(t)$$

Since the objective is purely maximization over a set of known positive facts ($\mathcal{C}_q$) and oracle-suggested positive facts ($\Omega_q$), and there are no negative samples (or margin/regularization terms that push away incorrect entities), this optimization procedure is fundamentally unstable and prone to collapse or severe overfitting during the inference-time fine-tuning.

The gradient updates aim to maximally increase the similarity (score $f(t)$) between the refined anchor embedding $s$ and all associated target embeddings ($o$ in $\mathcal{C}_q$ and $o_j$ in $\Omega_q$). If the number of refinement steps $T$ is too large, the embeddings could converge to a state where $s$ and all related target entities are pushed maximally close together. This would result in poor discrimination against other entities in the graph (i.e., non-target entities not included in $\Omega_q$) when the final ranking is performed.

The authors' defense that "our approach does not require any negatives, thus circumventing the problem of their synthetic generation" is not a justification for stability. The inclusion of a stability mechanism (like a margin, or pushing away local negatives) is essential so as to prevent convergence to trivial solution (that is move all embeddings into the same exact location).

The paper needs a detailed discussion or an ablation study showing how the parameters for training (e.g., number of epochs, and learning rate) affects the stability of the optimized embeddings.

## Weakness 2

The reliance on LLMs as an oracle to generate the set of additional introduces critical source of bias and limitations. The problem is that LLM is used as Filter for finding candidate queries.  The paper justifies the oracle by needing to enhance the small amount of knowledge available for low-degree entities. However, by selecting the top-$m$ entities closest to a barycenter, the LLM effectively acts as a gatekeeper for what constitutes a plausible answer. Any correct, low-degree answer that the misses will be excluded from $\Omega_q$ and therefore will not benefit from the refinement process. Instead of allowing the KG structure to guide the prediction and merely correcting the embedding landscape, the method constrains the space of beneficial answers to those already deemed similar or plausible by the LLM. This is a significant limitation, especially if the KG is used to "complement" the LLMs.

The lack of specific technical details on the LLM's performance (e.g., precision/recall of the generated triples $\Omega_q$ against the actual test set) makes it difficult to assess the quality of this filtering step.

## Weakness 3

The primary evidence (Figure 2) on the degree imbalance issue (performance systematically decreases as the normalized degree difference increases) is not clearly monotonic and appears to be heavily dependent on both the dataset and the underlying KGE model. No quantitative analysis is performed on this statement. While the overall trend is that the highest imbalance bins (e.g., the far left and far right) have the lowest MRR (on either object or subject corruption), there are significant counter-examples for FB15k-237, this trend did not hold true for YAGO3-10 (bins in far right, for example).  The lack of alignment in the patterns imply that there are different mechanisms at play that explain the low performance of these queries.

**Questions:**

1. What happens when we increate the number of iterations (W1)
2. How sensitive the proposed solution is to the LLM's knowledge. What if we have queries that are imbalanced and LLMs do not know.

---

> ### Author Response · Authors · 2025-12-04
> **Review answer**
>
> Thank you for your feedback. You raised valid points that we address here and in the updated version of the paper.
>
> **Weakness 1**: your concern on the stability of the optimization is very reasonable, and one we also expected to observe. However, we experimentally found that no collapse happen. We have added a paragraph in appendix D (Degenerate Behaviour Loss Function).
>
> **Weakness 2**: it is true that, if an entity is not included in the triples generated by the oracle, its embedding is not going to be touched by the optimization. However, in these cases, we rely on the refined representation of the anchor to help correctly answering the query:
>
> •	the first term of the loss pulls the embedding of the anchor in an area of the space to satisfy training context queries, thus allowing the KG structure to guide the prediction;
>
> •	the second term of the loss function, on the other hand, pulls the embedding of the anchor in an area of the space that satisfies triples deemed relevant by the oracle. The target of these triples could be similar (in the KGE embedding space) to the correct answer, even though such an answer was not included by the oracle (i.e., it was not similar in the LLM embedding space). Thus, the anchor is still moved closer to the correct answer even though the embedding of such answer is not refined.
>
> Referring to the example of the paper, the LLM might believe "k-pop", "poprock" and "europop" are more similar to "mandopop" (the target entity in the training context) than "j-pop" (the correct answer), that is thus not observed during the refinement. However, if k-pop and j-pop have similar KGE representations, the presence of "k-pop" among the entities suggested by the LLM is enough to pull the embedding of the anchor toward that region of the space, thus making "j-pop" a more compelling answer as well.
>
>
> **Weakness 3**: The interdependence between nodes, predicates, and topology that characterizes KGE model is so elevated that fully disentangling all these aspects is not possible. Therefore, our work identifies an issue that is broader and more precisely scoped than others identified before, but it can't be a complete solution to all issues such models present, which leaves some wiggle room for counterexamples.
>
> In addition, we observed that a dependency on the degree distribution of the knowledge graph can contribute: this is much more skewed on a dataset like FB15k-237 compared to Yago, which biases much more the learnt representation of the entities and thus makes degree imbalance a more pressing issue and the correlation more significant.
> Finally, the dependence on the KGE model reflects how different scoring functions capture some relation types better than others: for example, ComplEx poorly models composition, while RotatE better supports it. This will inevitably impact the way the two models might perform on the same queries, independently on the degree imbalance.
>
> Question 1: addressed in Weakness one.
>
> Question 2: addressed in the general comment for the AC.

---

### Note · Authors · 2026-01-12

**Comment:**

Opted for withdrawal as chances of acceptance are almost null and waiting for the official response would have prevented us from resubmitting this work to a different venue.

**Withdrawal Confirmation:**

I have read and agree with the venue's withdrawal policy on behalf of myself and my co-authors.